# FLatS: Principled Out-of-Distribution Detection with Feature-Based Likelihood Ratio Score

**Haowei Lin**[1,2,3*]    **Yuntian Gu**[3]

[1]Institute for Artificial Intelligence, Peking University
[2]School of Intelligence Science and Technology, Peking University
[3]Yuanpei College, Peking University
linhaowei@pku.edu.cn    guyuntian@stu.pku.edu.cn

## Abstract

Detecting out-of-distribution (OOD) instances is crucial for NLP models in practical applications. Although numerous OOD detection methods exist, most of them are empirical. Backed by theoretical analysis, this paper advocates for the measurement of the "OOD-ness" of a test case $x$ through the *likelihood ratio* between out-distribution $\mathcal{P}_{out}$ and in-distribution $\mathcal{P}_{in}$. We argue that the state-of-the-art (SOTA) feature-based OOD detection methods, such as Maha (Lee et al., 2018) and KNN (Sun et al., 2022), are suboptimal since they only estimate in-distribution density $p_{in}(x)$. To address this issue, we propose **FLatS**, a principled solution for OOD detection based on likelihood ratio. Moreover, we demonstrate that FLatS can serve as a general framework capable of enhancing other OOD detection methods by incorporating out-distribution density $p_{out}(x)$ estimation. Experiments show that FLatS establishes a new SOTA on popular benchmarks.[1]

## 1 Introduction

Natural language processing systems deployed in real-world scenarios frequently encounter out-of-distribution (OOD) instances that fall outside the training corpus distribution. For instance, it is hard to cover all potential user intents during the training of a task-oriented dialogue model. Therefore, it becomes crucial for practical systems to detect these OOD intents or classes during the testing phase. The ability to detect OOD instances enables appropriate future handling, including additional labeling and utilization for system updates, ensuring the system's continued improvement (Ke et al., 2022, 2023).

A rich line of work has been proposed to tackle OOD detection. Among them, the best-performing methods exploit the information of feature / hidden

---

*Corresponding author.
[1]Our code is publicly available at https://github.com/linhaowei1/FLatS.

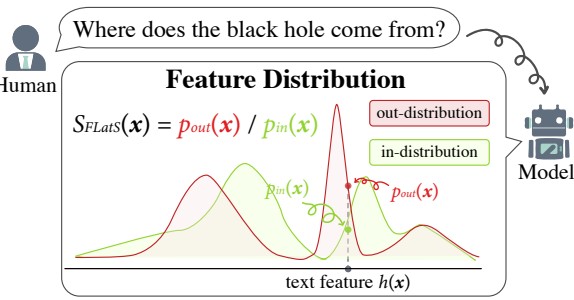

Figure 1: The framework of OOD detection with feature-based likelihood ratio score (FLatS). The model extracts the feature of input text, and then outputs the OOD score $S_{FLatS}(x)$ that takes the form of likelihood ratio between out-distribution $\mathcal{P}_{out}$ and in-distribution $\mathcal{P}_{in}$.

representation $h(x; \theta)$ of test case $x$ encoded by the tested NLP model. For example, Maha (Lee et al., 2018) estimates the Mahalanobis distance between $h(x; \theta)$ to the in-distribution (IND), while KNN (Sun et al., 2022) estimates the distance to the $k$-nearest IND neighbor. These techniques have demonstrated remarkable performance in recent benchmark studies (Yang et al., 2022; Zhang et al., 2023).

However, these methods were proposed without principled guidance. To address this, our paper first formulates OOD detection as a binary hypothesis test problem and derives that the principled solution towards OOD detection is to estimate the likelihood ratio $p_{out}(x)/p_{in}(x)$. Under this framework, we show that Maha and KNN only estimates IND density $p_{in}(x)$ and assumes OOD distribution $\mathcal{P}_{out}$ to be *uniform distribution*, which is sub-optimal. This paper then proposes a principled solution for OOD detection with feature-based likelihood ratio score, namely **FLatS**. In FLatS, the IND density $p_{in}(x)$ is also estimated with KNN on the training corpus, while the OOD density $p_{out}(x)$ is estimated with KNN on OOD data. Though we are not access to the real OOD data, we leverage public corpus (e.g., Wiki, BookCorpus) as auxiliary OOD data. Apart

from KNN, we further demonstrate that the idea of FLatS to incorporate OOD distribution information is applicable to other OOD detection techniques. Experiments demonstrate the effectiveness of the proposed FLatS.

## 2 Background

This paper focuses on supervised multi-class classification, a widely studied setting in OOD detection. The formal definition is given as follows:

**Definition 1 (OOD detection)** *Given an input space $\mathcal{X} \subset \mathbb{R}^d$ and a label space $\mathcal{Y} = \{1, ..., K\}$, $\mathcal{P}_{\mathcal{X}\mathcal{Y}}$ is a joint in-distribution (IND) over $\mathcal{X} \times \mathcal{Y}$. Given a training set $\mathcal{D} = \{(\boldsymbol{x}_j, y_j)\}_{j=1}^n$ drawn i.i.d. from $\mathcal{P}_{\mathcal{X}\mathcal{Y}}$, OOD detection aims to decide whether a test case $\boldsymbol{x} \in \mathcal{X}$ is drawn from the IND data distribution $\mathcal{P}_{in}$ (the marginal IND distribution on $\mathcal{X}$) or some OOD data distribution $\mathcal{P}_{out}$.*

OOD detection has been studied extensively. For example, using the *maximum softmax probability* (MSP) (Hendrycks and Gimpel, 2016) to measure IND-ness is popular in literature. There are more advanced methods like *maximum logit* (Hendrycks et al., 2019) and *energy score* (Liu et al., 2020).

Among the existing OOD detection methods, *distance-based* Mahalanobis (Maha) score and K-nearest neighbor (KNN) score achieve remarkable performance on common OOD detection benchmarks. These methods first extract latent feature $\boldsymbol{z} = h(\boldsymbol{x}; \theta)$ of test case $\boldsymbol{x}$ with the pre-trained language model $\theta$. For Maha and KNN, the OOD-ness of $\boldsymbol{x}$ are measured by the two scores[2]:

$$S_{Maha}(\boldsymbol{x}) = \min_{c \in \mathcal{Y}} (\boldsymbol{z} - \boldsymbol{\mu}_c)^T \boldsymbol{\Sigma}^{-1} (\boldsymbol{z} - \boldsymbol{\mu}_c), \quad (1)$$

$$S_{KNN}(\boldsymbol{x}; \mathcal{D}) = ||\boldsymbol{z}^* - kNN(\boldsymbol{z}^*; \mathcal{D}^*)||_2. \quad (2)$$

In Equation (1), $\boldsymbol{\mu}_c$ is the class centroid for class $c$ and $\Sigma$ is the global covariance matrix, which are estimated on IND training corpus $\mathcal{D}$. In Equation (2), $|| \cdot ||_2$ is Euclidean norm, $\boldsymbol{z}^* = \boldsymbol{z}/||\boldsymbol{z}||_2$ denotes the normalized feature $\boldsymbol{z}$, and $\mathcal{D}^*$ denotes the set of normalized features from training set $\mathcal{D}$. $kNN(\boldsymbol{z}^*; \mathcal{D}^*)$ denotes the $k$-nearest neighbor of $\boldsymbol{z}^*$ in set $\mathcal{D}^*$. More details are given in Appendix B.

---

[2]Note that $S(\boldsymbol{x})$ in this paper measures **OOD-ness** of $\boldsymbol{x}$, which means OOD sample will have high $S(\boldsymbol{x})$. Many literature define $S(\boldsymbol{x})$ to measure the **IND-ness** of $\boldsymbol{x}$.

## 3 Method

### 3.1 A Principled Solution for OOD Detection

In his seminal work, Bishop (1994) framed OOD detection as a selection problem between the in-distribution $\mathcal{P}_{in}$ and an out-of-distribution $\mathcal{P}_{ood}$. From a frequentist perspective, the objective of OOD detection can be formulated as a binary hypothesis test (Zhang and Wischik, 2022):

$$\mathcal{H}_0 : \boldsymbol{x} \sim \mathcal{P}_{out} \quad v.s. \quad \mathcal{H}_1 : \boldsymbol{x} \sim \mathcal{P}_{in} \quad (3)$$

By leveraging the Neyman-Pearson lemma (Neyman and Pearson, 1933), Theorem 1 demonstrates that likelihood ratio is a principled solution for OOD detection (the proof is given in Appendix A):

**Theorem 1** *A test with rejection region $\mathcal{R}$ defined as follows is a unique **uniformly most powerful (UMP)** test for the test problem defined in Equation (3):*

$$\mathcal{R} := \{\boldsymbol{x} : p_{out}(\boldsymbol{x})/p_{in}(\boldsymbol{x}) < \lambda_0\},$$

*where $\lambda_0$ is a threshold that can be chosen to obtain a specified significance level.*

Theorem 1 highlights the importance of detecting OOD samples based on both low IND density $p_{in}(\boldsymbol{x})$ and high OOD density $p_{out}(\boldsymbol{x})$. However, most distance-based OOD detectors are basically probability density estimators that only estimate IND density $p_{in}(\boldsymbol{x})$ with training data, and assume OOD distribution $\mathcal{P}_{out}$ as uniform distribution (see Appendix B for justifications).

Assuming a uniform OOD distribution $\mathcal{P}_{out}$ may lead to potential risks. For instance, consider a scenario where $\mathcal{P}_{out} = \mathcal{N}(0, 0.01)$ and $\mathcal{P}_{in} = \mathcal{N}(0, 1)$. It is apparent that 0 has higher IND density than 1: $p_{in}(0) > p_{in}(1)$, but 0 is indeed more OOD-like than 1: $10 = p_{out}(0)/p_{in}(0) > p_{out}(1)/p_{in}(1) = 10 \cdot e^{-49.5}$. This toy case illustrates that OOD detection cannot be based solely on IND density but should incorporate both IND and OOD densities.

Although we derive the principled solution for OOD detection with likelihood ratio, it is noteworthy that we typically have no access to genuine OOD data in real application, thus the OOD density $p_{out}(\boldsymbol{x})$ is hard to estimate. To address this, we follow recent works (Xu et al., 2021) to make use of a public corpus (e.g., Wiki, BookCorpus (Zhu et al., 2015)) to serve as auxiliary OOD data.

## 3.2 Feature-based Likelihood Ratio Score

This subsection designs an OOD score based on the likelihood ratio $p_{out}(\boldsymbol{x})/p_{in}(\boldsymbol{x})$ as motivated by Theorem 1. Since it is challenging to directly estimate the raw data distribution within the high-dimensional *text space*, we consider estimation in the low-dimensional *feature space*. As Appendix B suggests, $S_{Maha}(\boldsymbol{x})$ and $S_{KNN}(\boldsymbol{x})$ defined in Equation (1) and Equation (2) essentially function as density estimators that estimate the IND distribution $\mathcal{P}_{in}$ in the feature space. We will also exploit them to estimate OOD distribution $\mathcal{P}_{out}$ in our proposed method.

To connect the normalized probability densities with unnormalized OOD scores, we leverage energy-based models (EBMs) to parameterize $\mathcal{P}_{in}$ and $\mathcal{P}_{out}$: Given a test case $\boldsymbol{x}$, it has density $p_{in}(\boldsymbol{x}) = \exp\{-E_{in}(\boldsymbol{x})\}/Z_1$ in $\mathcal{P}_{in}$, and density $p_{ood}(\boldsymbol{x}) = \exp\{-E_{out}(\boldsymbol{x})\}/Z_2$ in $\mathcal{P}_{out}$, where $Z_1, Z_2$ are noramlizing constants that ensure the integral of densities $p_{in}(\boldsymbol{x})$ and $p_{out}(\boldsymbol{x})$ equal 1, and $E_{in}(\cdot), E_{out}(\cdot)$ are called *energy functions*. Then we can derive the OOD scores in the form of likelihood ratio with energy functions: $S(\boldsymbol{x}) = \log(p_{out}(\boldsymbol{x})/p_{in}(\boldsymbol{x})) = E_{in}(\boldsymbol{x}) - E_{out}(\boldsymbol{x}) + \log(Z_2/Z_1)$. Since $\log(Z_2/Z_1)$ is a constant, it can be omitted in the OOD score definition:

$$S_{FLatS}(\boldsymbol{x}) = E_{in}(\boldsymbol{x}) - E_{out}(\boldsymbol{x}). \quad (4)$$

Since the energy function $E_{in}(\cdot)$ and $E_{out}(\cdot)$ do not need to be normalized, we can estimate them with OOD scores. For IND energy $E_{ind}(\boldsymbol{x})$, we simply adopt the OOD score $S_{KNN}(\boldsymbol{x})$. For OOD energy $E_{out}(\boldsymbol{x})$, we replace the training corpus $\mathcal{D}$ in Equation (2) with an auxiliary OOD corpus $\mathcal{D}_{aux}$:

$$S_{FLatS}(\boldsymbol{x}) = S_{KNN}(\boldsymbol{x}; \mathcal{D}) - \alpha \cdot S_{KNN}(\boldsymbol{x}; \mathcal{D}_{aux}). \quad (5)$$

Since $S_{KNN}(\boldsymbol{x}; \mathcal{D})$ and $S_{KNN}(\boldsymbol{x}; \mathcal{D}_{aux})$ may be in different scales, $\alpha$ is a scaling hyper-parameter to make the two scores comparable. To the best of our knowledge, this is the first feature-based OOD score that follows the principled likelihood ratio solution. Also, KNN in Equation (5) is only an example, which can be replaced by other feature-based OOD scores such as $S_{Maha}(\boldsymbol{x})$ (see Section 4.3 for ablation studies on different estimation methods).

## 4 Experimental Setup

### 4.1 Datasets and Baselines

**Datasets.** We utilize 4 intent classification datasets CLINC150 (Larson et al., 2019), ROSTD (Gangal et al., 2020), Banking77 (Casanueva et al., 2020), and Snips (Coucke et al., 2018) for our experiments, which are commonly used in OOD detection literature. For each dataset, we use some classes as IND and the remaining classes as OOD classes. More details can be found in Appendix C.

**Choice of auxiliary OOD corpus $\mathcal{D}_{aux}$.** We adopt English Wikipedia,[3] which is the source used in common by RoBERTa for pre-training.

**Baselines.** We compare the proposed FLatS with 9 popular OOD detection methods. (1) For *confidence-based methods* that leverages output probabilities of classifiers trained on IND data to detect OOD samples, we evaluate **MSP** (Lee et al., 2018), **energy score** (Liu et al., 2020), **ODIN** (Liang et al., 2017), **D2U** (Chen et al., 2023), **MLS** (Hendrycks et al., 2019); (2) For *distance-based methods*, we test **LOF** (Breunig et al., 2000), **Maha** (Lee et al., 2018), **KNN** (Sun et al., 2022), and **GNOME** (Chen et al., 2023).

**Evaluation Metrics.** We adopt two widely-used metrics AUROC and FPR@95 following prior works (Yang et al., 2022). Higher AUROC and lower FPR@95 indicate better performance.

### 4.2 Implementation Details

**Architecture.** We adopt RoBERTa$_{\textbf{BASE}}$ as our backbone model. The model is fine-tuned on IND training datasets before OOD detection evaluation. The fine-tuning follows the standard practice (Kenton and Toutanova, 2019), where we pass the final layer `` token representation to a feed-forward classifier with softmax output for label prediction, together trained with cross-entropy loss.

**Hyperparameters.** We use $k = 10$ for KNN following (Chen et al., 2023). Searching from $\{0.1, 0.2, 0.5, 1.0, 2.0\}$, we adopt $\alpha = 0.5$ for Equation (5). We use Adam optimizer with a learning rate of $2e - 5$, a batch size of 16 and 5 fine-tuning epochs. We evaluate the model on IND validation set after every epoch and choose the best checkpoint with the highest IND classification accuracy.

---

[3]https://dumps.wikimedia.org

| | CLINC150 | | ROSTD | | Banking77 | | Snips | |
|---|---|---|---|---|---|---|---|---|
| | AUROC ↑ | FPR@95 ↓ | AUROC ↑ | FPR@95 ↓ | AUROC ↑ | FPR@95 ↓ | AUROC ↑ | FPR@95 ↓ |
| MSP | $95.72^{\pm0.18}$ | $19.08^{\pm0.55}$ | $75.42^{\pm0.05}$ | $51.24^{\pm0.21}$ | $83.35^{\pm0.10}$ | $56.20^{\pm0.32}$ | $79.17^{\pm0.22}$ | $56.15^{\pm0.66}$ |
| Energy | $96.18^{\pm0.12}$ | $15.76^{\pm0.43}$ | $76.52^{\pm0.10}$ | $52.53^{\pm0.32}$ | $82.64^{\pm0.22}$ | $51.02^{\pm0.58}$ | $75.10^{\pm0.32}$ | $40.64^{\pm0.63}$ |
| ODIN | $96.20^{\pm0.11}$ | $15.90^{\pm0.42}$ | $75.71^{\pm0.09}$ | $52.15^{\pm0.33}$ | $83.05^{\pm0.24}$ | $50.74^{\pm0.59}$ | $80.65^{\pm0.31}$ | $51.34^{\pm0.61}$ |
| D2U | $96.26^{\pm0.15}$ | $15.66^{\pm0.45}$ | $75.72^{\pm0.11}$ | $52.14^{\pm0.39}$ | $83.08^{\pm0.21}$ | $50.19^{\pm0.55}$ | $80.65^{\pm0.36}$ | $51.33^{\pm0.66}$ |
| MLS | $96.36^{\pm0.13}$ | $16.40^{\pm0.44}$ | $76.54^{\pm0.11}$ | $52.35^{\pm0.33}$ | $82.62^{\pm0.22}$ | $50.65^{\pm0.58}$ | $75.11^{\pm0.32}$ | $40.65^{\pm0.64}$ |
| LOF | $97.17^{\pm0.10}$ | $14.58^{\pm0.45}$ | $97.49^{\pm0.05}$ | $4.69^{\pm0.23}$ | $92.73^{\pm0.12}$ | $41.49^{\pm0.25}$ | $94.13^{\pm0.21}$ | $13.37^{\pm0.54}$ |
| Maha | $97.57^{\pm0.09}$ | $12.26^{\pm0.43}$ | $99.66^{\pm0.04}$ | $1.06^{\pm0.21}$ | $92.64^{\pm0.15}$ | $41.54^{\pm0.31}$ | $94.33^{\pm0.18}$ | $13.82^{\pm0.58}$ |
| KNN | $97.53^{\pm0.11}$ | $13.50^{\pm0.45}$ | $99.67^{\pm0.03}$ | $0.71^{\pm0.18}$ | $92.74^{\pm0.15}$ | $42.04^{\pm0.22}$ | $94.44^{\pm0.19}$ | $13.38^{\pm0.54}$ |
| GNOME | $96.84^{\pm0.14}$ | $14.94^{\pm0.65}$ | $99.63^{\pm0.10}$ | $1.47^{\pm0.28}$ | $91.43^{\pm0.09}$ | $44.23^{\pm0.24}$ | $92.58^{\pm0.25}$ | $14.45^{\pm0.66}$ |
| **FLatS** | $\mathbf{97.80}^{\pm0.12}$ | $\mathbf{9.90}^{\pm0.65}$ | $\mathbf{99.83}^{\pm0.02}$ | $\mathbf{0.21}^{\pm0.03}$ | $\mathbf{93.85}^{\pm0.10}$ | $\mathbf{40.02}^{\pm0.23}$ | $\mathbf{97.98}^{\pm0.17}$ | $\mathbf{9.62}^{\pm0.63}$ |

Table 1: OOD detection performance (higher AUROC ↑ and lower FPR@95 ↓ is better) on the 4 benchmark datasets. All values are percentages averaged over 5 different random seeds, and the best results are highlighted in **bold**.

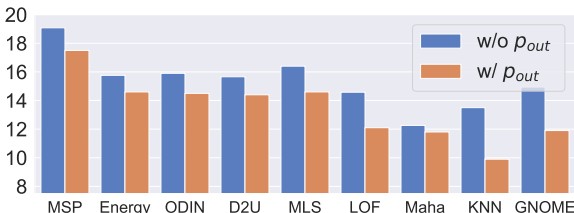

Figure 2: Ablation **Setting 1**: Average FPR@95 (%) for baselines on CLINC150 with (w/) or without (w/o) incorporation of OOD density estimation.

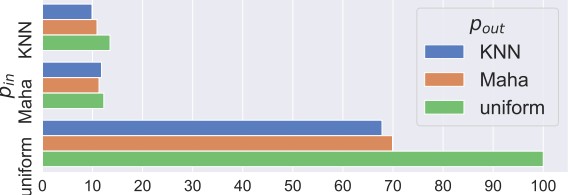

Figure 3: Ablation **Setting 2**: Average FPR@95 (%) for baselines on CLINC150 with different IND / OOD density estimation methods (uniform, Maha, KNN).

## 4.3 Ablation Settings

Note that $S_{FLatS}(\boldsymbol{x})$ in Equation (5) is only an illustrative method based on KNN. The concept of principled likelihood ratio can be extended within a broader framework to develop more OOD scores. To comprehensively assess the potential of this idea, we conduct two additional ablation studies:

**Setting 1:** In this setting, we aim to enhance the existing baselines by incorporating OOD density estimation. We replace $E_{in}(\boldsymbol{x})$ in Equation (4) with baseline OOD scores. Meanwhile, we maintain $E_{out}(\boldsymbol{x})$ as $S_{KNN}(\boldsymbol{x}; \mathcal{D}_{aux})$, thus exploring the impact of incorporating OOD density estimation on performance improvement.

**Setting 2:** In this setting, we aim to study the effects of different estimation methods for both OOD density $p_{out}(\boldsymbol{x})$ and IND density $p_{in}(\boldsymbol{x})$. Specifically, we replace $E_{in}(\boldsymbol{x})$ and $E_{out}(\boldsymbol{x})$ in Equation (4) with $S_{uniform}(\boldsymbol{x}) \equiv \text{const.}$, $S_{Maha}(\boldsymbol{x})$, and $S_{KNN}(\boldsymbol{x})$.

## 5 Results and Analysis

**FLatS establishes a new SOTA.** As shown in Table 1, FLatS achieves the best performance on the four benchmark datasets. The second best methods are KNN and Maha, whose average FPR@95 are 17.71% and 17.41%. They are higher than the average FPR@95 of FLatS (14.94%), which confirms the superiority of our proposed FLatS.

**FLatS enhances other baselines.** Figure 2 shows the FPR@95 results on CLINC150 under ablation setting 1. We observe that all the baselines achieve lower FPR@95 results by incorporating OOD density estimation. Therefore, FLatS is not only a single method, but can serve as a general framework to improve other SOTA OOD methods.

**FLatS can adopt different $E_{in}$ and $E_{out}$.** Figure 3 shows the FPR@95 results on CLINC150 with different ways (uniform, Maha, KNN) to estimate $\mathcal{P}_{in}$ and $\mathcal{P}_{out}$. The results reveal that the incorporation of OOD distribution estimation (no matter KNN or Maha) is beneficial compared to assuming $\mathcal{P}_{out}$ as a uniform distribution.

## 6 Related Work

OOD detection is crucial for NLP applications (Ryu et al., 2018; Borjali et al., 2021). In test time, the key difference of OOD detection methods is the OOD score design, which can be roughly categorized into two branches: *confidence-based methods* (Hendrycks and Gimpel, 2016; Liu et al., 2020; Hendrycks et al., 2019), and *distance-based methods* (Lee et al., 2018; Sun et al., 2022; Breunig et al., 2000). Some textual OOD detection methods (Arora et al., 2021) also exploit *perplexity* of auto-regressive language models (Arora et al.,

2021). Leveraging auxiliary OOD data (collected public corpus or synthesized OOD data) for training has been considered in literature (Xu et al., 2021; Wang et al., 2022). However, none of the works use auxiliary OOD data to estimate OOD distribution $\mathcal{P}_{out}$, which is a key novelty of our paper. More related work can be found in this excellent survey (Lang et al., 2023).

There are also some previous works (Ren et al., 2019; Xiao et al., 2020) that use "likelihood ratio" to detect OOD samples. However, our FLatS framework is very different from these works in the following aspects: (1) They used *probabilistic generative models*, e.g., VAEs (Kingma and Welling, 2013), to estimate likelihood, which is hard to train and difficult to scale up (in visual domains), and less effective in OOD detection. (2) The "likelihood ratio" they used is not between $\mathcal{P}_{out}$ and $\mathcal{P}_{in}$, and thus neither of them is a principled OOD detection method. For example, Ren et al. (2019) exploits a "background generative model" trained using random perturbation and Xiao et al. (2020) leverages a variational posterior distribution for test samples. They can also be viewed as special cases of FLatS which are estimated with different proxy distributions.

## 7   Discussion

In the derivation of our FLatS framework, We exploit the energy-based models (EBMs) for parameterization. EBMs are known for their **flexibility** with sacrifice to their **tractability**. But in our case, we leverage their flexibility to derive principled OOD scores (following Theorem 1) while keep the tractability via approximation with traditional OOD scores (e.g., KNN) in real-world applications. The detailed explanation is shown as follows.

**Flexibility**: Since Theorem 1 suggests that we should design OOD scores under the form of likelihood ratio between $\mathcal{P}_{out}$ and $\mathcal{P}_{in}$, we adopt EBMs to model the two probability distributions $\mathcal{P}_{out}$ and $\mathcal{P}_{in}$ due to the flexibility of EBMs. Thanks to EBMs, we transform the computation of likelihood-ratio into two unnormalized energy functions $E_{out}(\boldsymbol{x})$ and $E_{in}(\boldsymbol{x})$ as shown in Section 3.2.

**Tractability**: Contrary to the traditional works that directly optimize EBMs via MCMC (Grathwohl et al., 2019; Lafon et al., 2023) which may face the problem of computational inefficiency, we approximate the energy functions using traditional feature-based OOD scores (KNN or Maha). The

efficiency of KNN in real-world applications has been proved in previous works (Ming et al., 2022; Yang et al., 2022). Therefore, our method FLatS that adopts KNN is scalable and efficient in real-world applications.

Also, though we use public corpus for the estimation of $\mathcal{P}_{out}$ in the experiments, FLatS is compatible with any desired OOD data when they are available. As FLatS does not require model re-training, it has great potential in test-time adaptation to tackle distribution shifting in real-world scenarios.

## 8   Conclusion

This paper proposes to solve OOD detection with feature-based likelihood ratio score, which is principled (justified by Theorem 1). The proposed FLatS is simple and effective, which not only establishes a new SOTA, but can serve as a general framework to improve other OOD detection methods.

## Limitations

We list two limitations of this work. First, this paper mainly focuses on the **post-hoc** OOD detection approaches. Post-hoc OOD detection methods compute the OOD score without any special training-time regularization for models. Although a large group of OOD detection methods are post-hoc, there are also some regularized fine-tuning schemes to improve the OOD detection capability of NLP models. Since FLatS can enhance other post-hoc OOD detection baselines as shown in Section 5, it is exciting to see if our FLatS can also improve those training-time techniques in the future work. Second, our proposed FLatS is based on the existing OOD score (KNN), and we do not propose any new score with novel estimation techniques. The main contribution of this paper is to solve OOD detection with principled likelihood ratio, and we will see if better scores can be developed to further improve the likelihood ratio estimation in the future.

## Ethics Statement

Since this research involves only classification of the existing datasets which are downloaded from the public domain, we do not see any direct ethical issue of this work. In this work, we provide a theoretically principled framework to solve OOD detection in NLP, and we believe this study will lead to intellectual merits that benefit from a reliable application of NLU models.

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

# A  Theoretical Analysis of Theorem 1

## A.1  Preliminary

### Definition 2 (Statistical hypothesis testing)

*Consider testing a null hypothesis $H_0 : \theta \in \Theta_0$ against an alternative hypothesis $H_1 : \theta \in \Theta_1$, where $\Theta_0$ and $\Theta_1$ are subsets of the parameter space $\Theta$ and $\Theta_0 \cap \Theta_1 = \emptyset$. A test consists of a test statistic $T(X)$, which is a function of the data $\boldsymbol{x}$, and a rejection region $\mathcal{R}$, which is a subset of the range of $T$. If the observed value $t$ of $T$ falls in $\mathcal{R}$, we reject $H_0$.*

### Definition 3 (UMP test)
*Denote the power function $\beta_{\mathcal{R}}(\theta) = P_\theta(T(x) \in \mathcal{R})$, where $P_\theta$ denotes the probability measure when $\theta$ is the true parameter. A test with a test statistic $T$ and rejection region $\mathcal{R}$ is called a **uniformly most powerful** (UMP) test at significance level $\alpha$ if it satisfies two conditions:*

1. *$\sup_{\theta \in \Theta_0} \beta_{\mathcal{R}}(\theta) \leq \alpha$.*

2. *$\forall \theta \in \Theta_1, \beta_{\mathcal{R}}(\theta) \geq \beta_{R'}(\theta)$ for every other test $T'$ with rejection region $R'$ satisfying the first condition.*

## A.2  Proof of Theorem 1

**Lemma 1** *(Neyman and Pearson, 1933)  Let $\{X_1, X_2, ..., X_n\}$ be a random sample with likelihood function $L(\theta)$. The UMP test of the simple hypothesis $H_0 : \theta = \theta_0$ against the simple hypothetis $H_a : \theta = \theta_a$ at level $\alpha$ has a rejection region of the form:*

$$\frac{L(\theta_0)}{L(\theta_a)} < k$$

*where $k$ is chosen so that the probability of a type I error is $\alpha$.*

Now the proof of Theorem 1 is straightforward. From Lemma 1, the UMP test for Equation (3) has a rejection region of the form:

$$\frac{p_{out}(\boldsymbol{x})}{p_{in}(\boldsymbol{x})} < \lambda_0$$

where $\lambda_0$ is is chosen so that the probability of a type I error is $\alpha$.

## A.3  UMP test achieves optimal AUROC

In this section, we show that the UMP test for Equation (3) also achieves the optimal AUROC, which is a popular metric used in OOD detection. From the definition of AUROC, we have:

$$
\begin{aligned}
AUROC &= \int_0^1 1 - FPR \, d(TPR) \\
&= \int_0^1 \beta_{\mathcal{R}}(\theta_{in}) d(1 - \beta_{\mathcal{R}}(\theta_{out})) \\
&= \int_0^1 \beta_{\mathcal{R}}(\theta_{in}) d(\beta_{\mathcal{R}}(\theta_{out})),
\end{aligned}
$$

where FPR and TPR are *false positive rate* and *true positive rate*. Therefore, an optimal AUROC requires UMP test of any given level $\alpha = \beta_{\mathcal{R}}(\theta_{out})$ except on a null set.

# B  Distance-Based OOD Detectors are IND Density Estimators

In this section, we will show that $S_{Maha}(\boldsymbol{x})$ and $S_{KNN}(\boldsymbol{x})$ defined in Equation (1) and Equation (2) are IND density estimators under different assumptions. Assume we have a feature encoder $\phi : \mathcal{X} \rightarrow \mathcal{R}^m$, and in training time we empirically observe $n$ IND samples $\{\phi(\boldsymbol{x}_1), \phi(\boldsymbol{x}_2)...\phi(\boldsymbol{x}_n)\}$.

**Analysis of Maha score.**  Denote $\boldsymbol{\Sigma}$ to be the covariance matrix of $\phi(\boldsymbol{x})$. The final feature we extract from data $\boldsymbol{z}$ is:

$$\boldsymbol{z}(\boldsymbol{x}) = A^{-1}\phi(\boldsymbol{x})$$

where $AA^T = \Sigma$. Note that the covariance of $\boldsymbol{z}$ is $\mathcal{I}$.

Given a class label $c$, we assume the distribution $z(x|c)$ follows a gaussian $\mathcal{N}(A^{-1}\mu_c, \mathcal{I})$. Immediately we have $\boldsymbol{\mu}_c$ to be the class centroid for class $c$ under the maximum likelihood estimation. We can now clearly address the relation between Maha score and IND density:

$$S_{Maha}(\boldsymbol{x}) = -2\max_{c\in\mathcal{Y}}(\ln p_{in}(\boldsymbol{x}|c)) - m\ln 2\pi$$

**Analysis of KNN score.**  We use normalized feature $\boldsymbol{z}(x) = \phi(\boldsymbol{x})/||\phi(\boldsymbol{x})||_2$ for OOD detection. The probability function can be attained by:

$$p_{in}(\boldsymbol{z}) = \lim_{r\to 0}\frac{p(\boldsymbol{z}' \in B(\boldsymbol{z}, r))}{|B(z,r)|}$$

where $B(\boldsymbol{z}, r) = \{\boldsymbol{z}' : ||\boldsymbol{z}' - \boldsymbol{z}||_2 \leq r \wedge ||\boldsymbol{z}'|| = 1\}$
Assuming each sample $\boldsymbol{z}(\boldsymbol{x}_i)$ is $i.i.d$ with a probability mass $1/n$, the density can be estimated by k-NN distance. Specifically, $r = ||\boldsymbol{z} - kNN(\boldsymbol{z})||_2$, $p(\boldsymbol{z}' \in B(\boldsymbol{z}, r)) = k/n$ and $|B(\boldsymbol{z}, r)| = \frac{\pi^{(m-1)/2}}{\Gamma(\frac{m-1}{2} + 1)}r^{m-1} + o(r^{m-1})$, where $\Gamma$ is Euler's gamma function. When $n$ is large and $k/n$ is small, we have the following equations:

$$p_{in}(\boldsymbol{x}) \approx \frac{k\Gamma(\frac{m-1}{2} + 1)}{\pi^{(m-1)/2}nr^{m-1}}$$

$$S_{KNN}(\boldsymbol{x}) \approx (\frac{k\Gamma(\frac{m-1}{2} + 1)}{\pi^{(m-1)/2}n})^{\frac{1}{m-1}}(p_{in}(\boldsymbol{x}))^{-\frac{1}{m-1}}$$

# C  Datasets Details

We use four publicly available intent classification datasets as benchmark datasets:

**CLINC150** (Larson et al., 2019) is a dataset specifically designed for OOD intent detection. It comprises 150 individual intent classes from diverse domains. The dataset contains a total of 22,500 IND queries and 1,200 OOD queries. The IND data is split into three subsets: 15,000 for training, 3,000 for validation, and 4,500 for testing. Additionally, the dataset includes 1,000 carefully curated OOD test data for evaluating performance on out-of-domain queries.

**ROSTD** (Gangal et al., 2020) is a large-scale intent classification dataset comprising 43,000 intents distributed across 13 intent classes. The dataset also includes carefully curated OOD intents. Following the dataset split, we obtain 30,521 samples for IND training, 4,181 samples for IND validation, 8,621 samples for IND testing, and 3,090 samples for OOD testing.

**Banking77** (Casanueva et al., 2020) is a fine-grained intent classification dataset focused on the banking domain. It consists of 9,003 user queries in the training set, 1,000 queries in the validation set, and 3,080 queries in the test set. The dataset encompasses 77 intent classes, of which 50 classes are used as IND classes, while the remaining 22 classes are designated as OOD classes.

**Snips** (Coucke et al., 2018) is a dataset containing annotated utterances gathered from diverse domains. Each utterance is assigned an intent label such as "Rate Book", "Play Music", or "Get Weather." The dataset encompasses 7 intent classes, of which 5 classes are used as IND classes, and the remaining 2 classes are used as OOD classes. After splitting the dataset, we obtain 9,361 IND training samples, 500 IND validation samples, 513 IND test samples, and 187 OOD test samples.

# D  Hardware and Software

We run all the experiments on NVIDIA GeForce RTX-2080Ti GPU. Our implementations are based on Ubuntu Linux 16.04 with Python 3.6.