# OpenReview forum: "FLatS: Principled Out-of-Distribution Detection with Feature-Based Likelihood Ratio Score"
_EMNLP/2023/Conference — EMNLP 2023 Main_

### Official Review · Reviewer_4HcV · 2023-07-29

**Soundness:** 4

**Excitement:**

4: Strong: This paper deepens the understanding of some phenomenon or lowers the barriers to an existing research direction.

**Paper Topic And Main Contributions:**

This paper proposes FLATS, a principled method for OOD detection based on likelihood ratio. FLATS is shown to enhance other OOD detection methods by incorporating out-distribution density estimation.

**Reasons To Accept:**

1. Backed by theoretical analysis, this paper introduce a novel principled OOD detection method FLATS that measures OOD-ness of an input through the likelihood ratio between OOD distribution and IND distribution.

2. Moreover, FLATS can serve as a general framework which is able to enhance other OOD detection methods by considering OOD density estimation.

3. A new SOTA on popular benchmarks is established. And the experiments and analysis are comprehensive and convincing.

**Reasons To Reject:**

1. The relation between this work and "Likelihood Ratios for Out-of-Distribution Detection" should be discussed. Likelihood ratio is used in both of these works.

**Reproducibility:**

4: Could mostly reproduce the results, but there may be some variation because of sample variance or minor variations in their interpretation of the protocol or method.

**Reviewer Confidence:**

4: Quite sure. I tried to check the important points carefully. It's unlikely, though conceivable, that I missed something that should affect my ratings.

---

> ### Author Rebuttal · Authors · 2023-08-27
>
> Dear Reviewer 4HcV,
>
> We sincerely appreciate the time and effort you have devoted to evaluating our paper! We are grateful for the constructive feedback provided and would like to address the concerns raised in the review as follows.
>
> > The relation between this work and "Likelihood Ratios for Out-of-Distribution Detection" should be discussed. Likelihood ratio is used in both of these works.
>
> Thanks for this advice! We acknowledge that it's important to discuss the relation between our paper and "Likelihood ratios for OOD detection" to improve clarity of our contributions and avoid potential misunderstandings.
>
> Firstly, there are some prior works [1-2] that use "likelihood ratios" to detect OOD samples. These works shared some similarity with ours:
>
> 1. They also noticed that the likelihood of IND distribution alone is not a good OOD score.
>
> 2. They also adopted "likelihood ratio" to address it, though the likelihood ratio is not between $\mathcal P_{out}$ and $\mathcal P_{in}$ as we did.
>
> However, our paper is very different from these works in the following aspects:
>
> 1. These works used **probabilistic generative models** to estimate likelihood, which is hard to train and difficult to scale up (in visual domains), and less effective in OOD detection.
> 2. The "likelihood ratio" they used is not between $\mathcal P_{out}$ and $\mathcal P_{in}$, and thus **neither of them is a principled OOD detection method**. For example, [1] exploits a "background generative model" trained using random perturbation and [2] leverages a variational posterior distribution for test samples. They can also be viewed as special cases of FLATS where $\mathcal P_{out}$ is estimated with different proxy distributions.
>
> We will add this discussion in our revised paper. We genuinely appreciate your feedback, which has prompted us to develop a more comprehensive analysis of FLATS' comparative strengths, thereby elevating the overall quality and depth of our revised manuscript.
>
> [1] Ren, Jie, et al. "Likelihood ratios for out-of-distribution detection." (NeurIPS 2019)
>
> [2] Xiao, Zhisheng, et al. "Likelihood regret: An out-of-distribution detection score for variational auto-encoder." (NeurIPS 2020)

---

### Official Review · Reviewer_RYgf · 2023-08-03

**Soundness:** 3

**Excitement:**

3: Ambivalent: It has merits (e.g., it reports state-of-the-art results, the idea is nice), but there are key weaknesses (e.g., it describes incremental work), and it can significantly benefit from another round of revision. However, I won't object to accepting it if my co-reviewers champion it.

**Paper Topic And Main Contributions:**

topic: detecting out-of-distribution (OOD) instances in NLP models
main contribution: they think the Sota feather-based OOD detection methods such as MAha, and KNN is suboptimal since they only estimate in-distribution density, to address this issue, they propose FLATS, which can incorporate out-distribution density.

**Reasons To Accept:**

 they propose FLATS, which can incorporate out-distribution density.

**Reasons To Reject:**

1) The rationale behind opting for RoBERTa, as opposed to other language models with varying parameters like T5-3b, 11b, llama-3b, and 7b. It is crucial to present a comprehensive comparison with these different language models.
2) In Section 5, it is recommended to provide a short explanation for the superiority of this method.


**Reproducibility:**

4: Could mostly reproduce the results, but there may be some variation because of sample variance or minor variations in their interpretation of the protocol or method.

**Reviewer Confidence:**

3: Pretty sure, but there's a chance I missed something. Although I have a good feel for this area in general, I did not carefully check the paper's details, e.g., the math, experimental design, or novelty.

---

> ### Author Rebuttal · Authors · 2023-08-27
>
> Dear Reviewer RYgf,
>
> We sincerely appreciate the time and effort you have devoted to evaluating our paper! We are grateful for the constructive feedback provided and would like to address the concerns raised in the review as follows.
>
> > The rationale behind opting for RoBERTa, as opposed to other language models with varying parameters like T5-3b, 11b, llama-3b, and 7b. It is crucial to present a comprehensive comparison with these different language models.
>
> Thanks for this suggestion! We also tried other backbones to validate our proposed FLATS approach and the results (AUROC / FPR@95) are shown as follows.
>
> |            | Bert-base  | Bert-large | Roberta-base        | Roberta-large       | T5-3b (LoRA)        | T5-11b (LoRA)       | llama-3b (LoRA)     | llama-7b (LoRA)     |
> | ---------- | ------------------- | ------------------- | ------------------- | ------------------- | ------------------- | ------------------- | ------------------- | ------------------- |
> | #params | 110M                | 340M                | 125M                | 335M                | 3B                  | 11B                 | 3B                  | 7B                  |
> | KNN        | 97.3 / 13.5         | 98.0 / 8.00         | 97.5 / 13.5         | 98.1 / 7.50         | 98.1 / 8.90         | 98.3 / 7.01         | 97.0 / 13.6         | 97.5 / 12.3         |
> | FLATS      | **97.6** / **9.70** | **98.2** / **6.70** | **97.8** / **9.90** | **98.3** / **6.50** | **98.4** / **6.45** | **98.8** / **5.90** | **97.8** / **9.90** | **98.2** / **7.15** |
>
> **Table 1. The AUROC$\uparrow$ (left) and FPR@95$\downarrow$ (right) results of KNN and FLATS on CLINC150 with different backbones.**
>
> Here we include Bert-base, Bert-large, Roberta-base, Roberta-large, T5-3b, T5-11b, LLaMA-3b, LLaMA-7b as the tested backbones. We conducted experiments on CLINC150 dataset as introduced in Appendix Section C. For LLMs with sizes larger than 3B, we adopt LoRA [1] to do parameter-efficient tuning to avoid out-of-memory issue. From the table we could see, **FLATS consistently improves the OOD detection performance across different backbones**, which demonstrate its empirical superiority.
>
> > In Section 5, it is recommended to provide a short explanation for the superiority of this method.
>
> Thanks for this advice. Apart from the quantitative analysis in section 5, we acknowledge that it's important to add more qualitative discussion. We will take this advice to add detailed analysis of the strengths of FLATS in our revised paper. Since we cannot edit the paper at this stage, we provide a brief summary of the superiority of FLATS as follows:
>
> - **Effectiveness:** Our approach capitalizes on feature-based information, a facet overlooked by both traditional and contemporary methods like MSP, Energy, ODIN, MLS. Research underscores the superiority of feature-based OOD detection techniques [2].
> - **Principled**. Traditional and contemporary feature-based methods KNN, Maha, LOF, GNOME are not principled as analyzed in line 123-128. FLATS is grounded in theoretical rigor, making it a more principled option.
> - **Flexibility:** While contemporaneous methods tend to be specialized and intricate, our FLATS framework is characterized by its simplicity and broad applicability, as evidenced in section 5. For example, GNOME devises to aggregate normalized scores produced by different models, and it has a trade-off between distribution shift types.
>
> We genuinely appreciate your feedback, which has prompted us to develop a more comprehensive analysis of FLATS' comparative strengths, thereby elevating the overall quality and depth of our revised manuscript.
>
> [1] Hu, Edward J., et al. "Lora: Low-rank adaptation of large language models." (ICLR 2021).
>
> [2] Yang, Jingkang, et al. "Openood: Benchmarking generalized out-of-distribution detection." (NeurIPS 2022)

---

### Official Review · Reviewer_RnjR · 2023-08-17

**Typos Grammar Style And Presentation Improvements:** no
**Soundness:** 4

**Excitement:**

4: Strong: This paper deepens the understanding of some phenomenon or lowers the barriers to an existing research direction.

**Missing References:**

no

**Paper Topic And Main Contributions:**

OOD detection is an important topic but most of the previous works are empirical. This paper demonstrates a theoretical understanding that uses the likelihood ratio between out-distribution and in-distribution to measure the extent of OOD. Based on this, the authors claim that SOTA methods are suboptimal because they only estimate the in-distribution density. The authors propose a method named FLATS, which outperform previous methods on several OOD detection benchmark.

**Questions For The Authors:**

1. Have you conducted experiments on other backbones or other than NLP classification tasks?

**Reasons To Accept:**

1. This paper is well-written and organized.
2. This paper has contributions from both theoretical and empirical perspectives: Not only provides a theoretical understanding and analyses of previous methods but also proposes a novel method that outperforms previous methods.

**Reasons To Reject:**

1. Experimental validation is weak, only considering Roberta as the backbone.

**Reproducibility:**

3: Could reproduce the results with some difficulty. The settings of parameters are underspecified or subjectively determined; the training/evaluation data are not widely available.

**Reviewer Confidence:**

2: Willing to defend my evaluation, but it is fairly likely that I missed some details, didn't understand some central points, or can't be sure about the novelty of the work.

---

> ### Author Rebuttal · Authors · 2023-08-27
>
> Dear Reviewer RnjR,
>
> We sincerely appreciate the time and effort you have devoted to evaluating our paper! We are grateful for the constructive feedback provided and would like to address the concerns raised in the review as follows.
>
> > Experimental validation is weak, only considering Roberta as the backbone.
>
> Thanks for this suggestion! We also tried other backbones to validate our proposed FLATS approach and the results (AUROC / FPR@95) are shown as follows.
>
> |            | Bert-base-uncased   | Bert-large-uncased  | Roberta-base        | Roberta-large       | T5-3b (LoRA)        | T5-11b (LoRA)       | LLaMA-3b (LoRA)     | LLaMA-7b (LoRA)     |
> | ---------- | ------------------- | ------------------- | ------------------- | ------------------- | ------------------- | ------------------- | ------------------- | ------------------- |
> | #parameter | 110M                | 340M                | 125M                | 335M                | 3B                  | 11B                 | 3B                  | 7B                  |
> | KNN        | 97.3 / 13.5         | 98.0 / 8.00         | 97.5 / 13.5         | 98.1 / 7.50         | 98.1 / 8.90         | 98.3 / 7.01         | 97.0 / 13.6         | 97.5 / 12.3         |
> | FLATS      | **97.6** / **9.70** | **98.2** / **6.70** | **97.8** / **9.90** | **98.3** / **6.50** | **98.4** / **6.45** | **98.8** / **5.90** | **97.8** / **9.90** | **98.2** / **7.15** |
>
> **Table 1. The AUROC$\uparrow$ (left) and FPR@95$\downarrow$ (right) results of KNN and FLATS on CLINC150 with different backbones.**
>
> Here we include Bert-base-uncased, Bert-large-uncased, Roberta-base, Roberta-large, T5-3b, T5-11b, LLaMA-3b, LLaMA-7b as the tested backbones. We conducted experiments on CLINC150 dataset as introduced in Appendix Section C. For LLMs with sizes larger than 3B, we adopt LoRA [1] to do parameter-efficient tuning to avoid out-of-memory issue. From the table we could see, **FLATS consistently improves the OOD detection performance across different backbones**, which demonstrate its empirical superiority.
>
> >  Have you conducted experiments on other backbones or other than NLP classification tasks?
>
> Yes. The experiments on other backbones are shown in the response to your last comment.
>
> We also tried NLP **non-classification** tasks and results are shown in **Table 2** below. While classification tasks are the most popular settings for OOD detection in NLP, we also noticed that there are works [2] focusing on non-classification tasks. In this setting, no class label is available and we can only train the model via unsupervised learning on IND data. We follow [2] to conduct experiments on SST and CLINC150 datasets with **Roberta-base** as the backbone. EDF, MDF, MDF (BCAD), MDF (IMLM), MDF(BCAD, IMLM) are methods proposed by [2]. Notice that other baselines in our paper are not suitable for non-classification settings, thus only KNN and FLATS are compared as they don't need class label in inference. To adopt KNN and FLATS to non-classification setting, the backbone is trained using unsupervised mask-language-model (MLM) loss on IND data.
>
> |                         | EDF  | MDF  | MDF (BCAD) | MDF (IMLM) | MDF (BCAD, IMLM) | KNN  | FLATS     |
> | ----------------------- | ---- | ---- | ---------- | ---------- | ---------------- | ---- | --------- |
> | SST (unsupervised)      | 99.5 | 99.8 | 99.2       | 99.9       | 99.9             | 99.7 | **100.0** |
> | CLINC150 (unsupervised) | 56.9 | 78.6 | 80.5       | 80.1       | 84.4             | 89.3 | **92.9**  |
>
> **Table 2. The AUROC$\uparrow$ results of different unsupervised OOD detection methods on SST and CLINC150 with Roberta-base.**
>
> From the table we can see, our FLATS also establishes a new state-of-the-art on non-classification NLP tasks. As non-classification tasks have become more and more popular in the LLM-era, we believe that the proposed FLATS has the potential to be widely adopted in many real-world applications.
>
> We genuinely appreciate your feedback, which has prompted us to develop a more comprehensive analysis of FLATS' comparative strengths, thereby elevating the overall quality and depth of our revised manuscript. If you have further questions, don't hesitate to contact us.
>
> [1] Hu, Edward J., et al. "Lora: Low-rank adaptation of large language models." (ICLR 2021).
>
> [2] Xu, Keyang, et al. "Unsupervised out-of-domain detection via pre-trained transformers." (ACL 2021).

---

### Official Review · Reviewer_UcFx · 2023-08-18

**Soundness:** 4

**Excitement:**

4: Strong: This paper deepens the understanding of some phenomenon or lowers the barriers to an existing research direction.

**Paper Topic And Main Contributions:**

This paper addresses the issue of detecting out-of-distribution (OOD) instances within the realm of Natural Language Processing (NLP) models. The paper's primary contributions are centered around the introduction of a novel methodology designed to assess the "OOD-ness" of a given test case. This is achieved through the application of a likelihood ratio, which juxtaposes the probabilities associated with out-of-distribution and in-distribution contexts. The paper posits that current state-of-the-art (SOTA) methods exhibit suboptimal performance due to their exclusive focus on estimating in-distribution density. In response, the paper presents a principled solution for OOD detection rooted in the concept of a likelihood ratio, denoted as FLATS. This framework holds the potential to serve as a versatile foundational structure, effectively bolstering various other OOD detection methodologies through the incorporation of out-distribution density.

**Reasons To Accept:**

1. The paper's strength lies in its rigorous approach to addressing out-of-distribution (OOD) detection, introducing the well-grounded feature-based likelihood ratio score substantiated by Theorem 1. The extension of FLATS to diverse feature-based OOD scores highlights its versatility and departure from existing methods, showing potential for real-world applications.

2. The experimental results underscore its contributions, as FLATS establishes itself as the new OOD detection state-of-the-art (SOTA), surpassing benchmarks like KNN and Maha. Experiments with various Pin and Pout estimation approaches further emphasize FLATS' adaptability and robustness, significantly boosting the paper's impact on the NLP community.

**Reasons To Reject:**

1. The method's reliance on likelihood ratio-based OOD scores, as justified by Theorem 1, warrants closer scrutiny. The application of energy-based models (EBMs) for parameterization introduces an inherent complexity, raising questions about the scalability and efficiency of the proposed framework, particularly in real-world applications.

2. The presented comparison with the KNN and Maha approaches is indicative, but a more detailed analysis of the strengths and weaknesses of FLATS in comparison to a broader spectrum of methods, including both traditional and contemporary alternatives, would contribute to a more nuanced understanding of its efficacy.

**Reproducibility:**

4: Could mostly reproduce the results, but there may be some variation because of sample variance or minor variations in their interpretation of the protocol or method.

**Reviewer Confidence:**

3: Pretty sure, but there's a chance I missed something. Although I have a good feel for this area in general, I did not carefully check the paper's details, e.g., the math, experimental design, or novelty.

---

> ### Author Rebuttal · Authors · 2023-08-27
>
> Dear Reviewer UcFx,
>
> We sincerely appreciate the time and effort you have devoted to evaluating our paper! We are grateful for the constructive feedback provided and would like to address the concerns raised in the review as follows.
>
> >  The method's reliance on likelihood ratio-based OOD scores, as justified by Theorem 1, warrants closer scrutiny.
>
> Thanks for your suggestion. Theorem 1 establishes the sound theoretical foundation of our approach, demonstrating its robustness. The theorem indicates that the likelihood ratio-based OOD scores are principled, and our empirical study further demonstrates the superiority. We will clarify on this to ensure a rigorous scrutiny in our revised paper.
>
> > The application of energy-based models (EBMs) for parameterization introduces an inherent complexity, raising questions about the scalability and efficiency of the proposed framework, particularly in real-world applications.
>
> We acknowledge the concern regarding the use of energy-based models (EBMs) for parameterization. EBMs are known for its **flexibility** with sacrifice to its **tractability**. But actually, we exploit its **flexibility** to derive principled OOD scores following Theorem 1 while keep the **tractability** via approximation with traditional OOD scores (e.g., KNN) in real-world applications. The detailed explanation is shown as follows.
>
> **Flexibility**: Since Theorem 1 suggests that we should design OOD scores under the form of likelihood ratio between $\mathcal P_{out}$ and $\mathcal P_{in}$, we adopt energy-based models (EBMs) to model the two probability distributions $\mathcal P_{out}$ and $\mathcal P_{in}$ due to the **flexibility** of EBMs. Thanks to EBMs, we transform the computation of likelihood-ratio $p_{out}(x)/p_{in}(x)$ into two unnormalized energy functions $E_{out}(x)$ and $E_{in}(x)$ (see line 158-174).
>
> **Tractability**: Contrary to the traditional works that directly optimize EBMs via MCMC [1, 2] which may face the problem of intractability, we approximate the energy functions using *traditional feature-based OOD scores* (KNN, Maha, or Uniform. see line 248-252). The efficiency of KNN in real-world applications has been proved in previous works [3, 4, 5]. Therefore, our method FLATS that adopts KNN is scalable and efficient in real-world applications.
>
> Thanks for this valuable feedback. We will include this discussion in our revised paper which will significantly improve the clarity and highlight the contribution of our paper.
>
> > The presented comparison with the KNN and Maha approaches is indicative, but a more detailed analysis of the strengths and weaknesses of FLATS in comparison to a broader spectrum of methods, including both traditional and contemporary alternatives, would contribute to a more nuanced understanding of its efficacy.
>
> We acknowledge the need for a more comprehensive analysis of FLATS against a wider array of methods. Apart from the quantitative analysis in section 5, it's important to add more qualitative discussion. We will take this advice to add detailed analysis of the strengths and weaknesses of FLATS in our revised paper. Since we cannot edit the paper at this stage, we provide a brief summary of the strengths and weaknesses as follows:
>
> **Strengths**:
>
> - **Effectiveness:** Our approach capitalizes on feature-based information, a facet overlooked by both traditional and contemporary methods like MSP, Energy, ODIN, MLS. Research underscores the superiority of feature-based OOD detection techniques [4].
> - **Principled**. Traditional and contemporary feature-based methods KNN, Maha, LOF, GNOME are not principled as analyzed in line 123-128. FLATS is grounded in theoretical rigor, making it a more principled option.
> - **Flexibility:** While contemporaneous methods tend to be specialized and intricate, our FLATS framework is characterized by its simplicity and broad applicability, as evidenced in section 5. For example, GNOME devises to aggregate normalized scores produced by different models, and it has a trade-off between distribution shift types.
>
> **weaknesses**:
>
> - **Post-hoc only**. It's worth noting that our OOD detection methodology operates in a post-hoc manner, without specific training-time regularization. We are intrigued by the potential for future work to explore integrating training-time techniques into the FLATS framework.
> - **Auxiliary Data Selection**. Some contemporary methods such as [6] designed advanced techniques to choose auxiliary OOD data. While currently FLATS did not elaborate on this, it's exciting to see if this will further enhance detection performance.
>
> [1] Lafon, Marc, et al. "Hybrid Energy Based Model in the Feature Space for Out-of-Distribution Detection." (ICML 2023)
>
> [2] Grathwohl, Will, et al. "Your classifier is secretly an energy based model and you should treat it like one." (ICLR 2020)
>
> [3] Sun, Yiyou, et al. "Out-of-distribution detection with deep nearest neighbors." (ICML 2022)
>
> [4] Yang, Jingkang, et al. "Openood: Benchmarking generalized out-of-distribution detection." (NeurIPS 2022)
>
> [5] Ming, Yifei, et al. "How to Exploit Hyperspherical Embeddings for Out-of-Distribution Detection?." (ICLR 2023)
>
> [6] Ming, Yifei, et al. "Poem: Out-of-distribution detection with posterior sampling." (ICML 2022)

---

### Meta-Review · Area_Chair_jQwe · 2023-09-19

**Recommendation:** 4

**Metareview:**

This papers tackles the problem of out-of-distribution detection. The reviewers appreciated the combination of theoretical foundations and empirical evaluations that outperform the proposed baselines on the given datasets. The authors were very responsive to the questions and feedback from the reviewers and the additional insights should certainly improve even more this paper.

---

### Decision · Program_Chairs · 2023-10-07

**Decision:**

Accept-Main

**Comment:**

This papers tackles the problem of out-of-distribution detection. The reviewers appreciated the combination of theoretical foundations and empirical evaluations that outperform the proposed baselines on the given datasets. The authors were very responsive to the questions and feedback from the reviewers and the additional insights should certainly improve even more this paper.